# PacBio long-read amplicon sequencing enables scalable high-resolution population allele typing of the complex *CYP2D6* locus

Sarah Charnaud[1,2,5], Jacob E. Munro [1,2,5], Lucie Semenec [1,2,3], Ramin Mazhari[1,2], Jessica Brewster[1,2], Caitlin Bourke [1,2], Shazia Ruybal-Pesántez [1,2,4], Robert James[1,2], Dulcie Lautu-Gumal[1,2], Harin Karunajeewa[1,2], Ivo Mueller [1,2,6] & Melanie Bahlo [1,2,6✉]

The CYP2D6 enzyme is estimated to metabolize 25% of commonly used pharmaceuticals and is of intense pharmacogenetic interest due to the polymorphic nature of the *CYP2D6* gene. Accurate allele typing of *CYP2D6* has proved challenging due to frequent copy number variants (CNVs) and paralogous pseudogenes. SNP-arrays, qPCR and short-read sequencing have been employed to interrogate *CYP2D6*, however these technologies are unable to capture longer range information. Long-read sequencing using the PacBio Single Molecule Real Time (SMRT) sequencing platform has yielded promising results for *CYP2D6* allele typing. However, previous studies have been limited in scale and have employed nascent data processing pipelines. We present a robust data processing pipeline "PLASTER" for accurate allele typing of SMRT sequenced amplicons. We demonstrate the pipeline by typing *CYP2D6* alleles in a large cohort of 377 Solomon Islanders. This pharmacogenetic method will improve drug safety and efficacy through screening prior to drug administration.

[1] Walter and Eliza Hall Institute of Medical Research, Melbourne, VIC, Australia. [2] Department of Medical Biology, University of Melbourne, Melbourne, VIC, Australia. [3] ARC Centre of Excellence in Synthetic Biology, Department of Molecular Sciences, Macquarie University, Sydney, NSW, Australia. [4] Burnet Institute, Melbourne, VIC, Australia. [5]These authors contributed equally: Sarah Charnaud, Jacob E. Munro. [6]These authors contributed equally: Ivo Mueller, Melanie Bahlo. ✉email: bahlo@wehi.edu.au

Pharmacogenetics is the study of the association between genetic variation and the differences in an individual's metabolic response to certain drugs. Genetic variation may modulate drug responses through alterations to both pharmacokinetic factors (i.e., what the body does to the drug) and pharmacodynamic factors (i.e., what the drug does to the body). Differences in pharmacogenetics can result in decreased drug efficacy, drug failure and even drug toxicity. It has been established that the frequencies of pharmacogenetic variants may vary widely between populations with different ancestries[1]. Consequently, the efficacy and safety of drug administration as reported in one population may be different from that of another population. Many clinical trials are biased towards participants of European ancestry, as are available genomic reference databases[2]. As such, pharmacogenetic screening of candidate populations prior to drug administration, particularly those of non-European ancestry, may provide valuable insights on how a given drug will perform.

An estimated 25% of all drugs are metabolized by a single enzyme, cytochrome P450-2D6 (CYP2D6)[3,4], which is highly expressed in the liver. CYP2D6 is highly polymorphic, with different alleles displaying a wide spectrum of enzymatic activity. As a simplification of the spectrum of CYP2D6 activity, an individual's metabolizer phenotype is classified into one of four categories: PM (poor metabolizer), IM (intermediate metabolizer), EM (normal or 'extensive' metabolizer), and UM (ultrarapid metabolizer)[5]. Estimates of metabolizer phenotype frequencies across several distinct populations, inferred from genotyping data, range from 0.4 to 5.4% for PMs, 0.4 to 11% for IMs, 67 to 90% for EMs, and 1 to 21% for UMs[6]. A key example of a pharmaceutical affected by CYP2D6 is the 8-aminoquinoline anti-malarial drug primaquine (PQ), which is a prodrug that requires metabolism by CYP2D6 for activity and is one of the only effective treatments for the prevention of Plasmodium vivax relapse. Patients with PM or IM phenotypes do not metabolize primaquine into its active metabolite[7] and are at higher risk of P. vivax relapse after treatment[8]. As such, knowledge of the frequency of PMs and IMs in a target population is critical when developing effective strategies for malaria control with PQ.

CYP2D6 alleles are composed of a diverse range of variant classes spanning single nucleotide polymorphisms (SNPs), insertions/deletions (indels), copy number variants (CNVs), structural variants (SVs), and gene fusions with the neighboring paralogous CYP2D7 pseudogene[9–11]. The PharmVar database, a centralized repository of pharmacogene variation, describes over 200 CYP2D6 alleles (www.pharmvar.org/gene/CYP2D6)[12]. Here, CYP2D6 alleles are named using the star (*) nomenclature system, wherein individual star alleles (e.g., *1, *2) are defined by the presence of a set of core variants that affect protein function. The additional variation that is not known or expected to affect protein function is captured in the so-called sub-alleles (e.g., *1.001 and *1.002 are sub-alleles of *1). To date, there are 126 CYP2D6 star alleles reported in PharmVar, covering 241 sub-alleles, of which 71 star alleles that cover 80 sub-alleles have unknown or uncertain function. Apart from the *5 deletion allele, CNVs are typically indicated with an "xN" suffix (e.g., *2xN) to indicate a duplication, or a hyphen to indicate a tandem arrangement of two distinct alleles (e.g., *36–*10[13]). In addition, fusion (hybrid) alleles resulting from a gene fusion event between CYP2D6 and CYP2D7 are collectively referred to as CYP2D6-D7 fusions when the 5′ portion is derived from CYP2D6 (e.g., *36), and CYP2D7-D6 when the 5′ portion is derived from CYP2D7 (*13 alleles)[11]. An effective approach for predicting CYP2D6 phenotype from genotype calls is through the use of a so-called "activity score", which is defined as the sum of the individual activity scores assigned to each allele present (scores of multicopy alleles are multiplied by the number of copies)[5]. An up-to-date database on CYP2D6 star allele activity scores is maintained on the PharmGKB website (https://www.pharmgkb.org/page/cyp2d6RefMaterials).

Commercial targeted genotyping assays based on microarrays, qPCR, or short-read sequencing have been applied as a relatively inexpensive approach to interrogate CYP2D6 variation[14–16]. Typically, these assays are limited in that they are only able to detect a predefined set of variants, allowing for the identification of only common CYP2D6 alleles. This limits the utility of these assays when considering uncharacterized populations, which may have unexpected allele frequencies and/or novel alleles. Short-read sequencing-based assays are able to discover some novel variation, but are confounded by the high sequence similarity between CYP2D6 and CYP2D7, which results in the misalignment of reads and erroneous variant calls[15]. In addition, phasing of variant calls to individual alleles cannot be achieved directly with these approaches, and instead is inferred based on the detected variants and known allele frequencies. As such, the presence of rare or novel alleles will further confound phasing attempts with these methods. The recently released short-read method Cyrius[17] is capable of accurately typing known CYP2D6 alleles from WGS data, but does not handle novel alleles which may be present in an uncharacterized population. In addition to the sequence of each CYP2D6 allele, the copy number state of each allele is also required to comprehensively predict CYP2D6 activity. This is typically derived from performing an additional qPCR assay that targets multiple CYP2D6 regions to identify both copy number and certain gene fusions with CYP2D7[18,19].

Long-read amplicon sequencing is capable of characterizing both known and novel alleles and offers a straightforward solution to the problem of variant phasing. However, the approach is not without its drawbacks. Long-read sequencing platforms are still maturing and typically suffer from higher per-base error rates compared to the commonly used short-read Illumina sequencing. Long-read sequencing error rates have been measured as 14.20% (dropping to 1.72% for circular consensus sequencing (CCS) with at least three passes) for PacBio and 20.19% (dropping to 13.41% for "2D" consensus sequencing) for Oxford Nanopore Technologies (ONT) reads[20], however more recent estimates factoring in platform improvements place this at <1% for PacBio CCS[21] and <5% for ONT[22]. The higher per-base error rate of long-read sequencing necessitates more rigorous data processing and increased read-depths to achieve similar levels of accuracy. A further complication arising in long-read amplicon typing is the formation of PCR chimeras during the amplification stage prior to sequencing, which increases with both the product length and the number of PCR cycles[23]. These chimeric products can be mistaken for true alleles, and as such, robust strategies for the chimera filtering need to be applied. Overall, the data processing pipeline for long-read amplicon sequencing is more demanding and methodologies are not yet as well established as for their short-read sequencing counterparts.

Long-read amplicon sequencing has previously been demonstrated to be highly informative for CYP2D6 typing using both the PacBio SMRT platform[24,25] and the ONT platform[26,27]. We have sought to build upon and improve previous work by: (1) designing a CYP2D6 amplicon sequencing strategy for use with field samples (2) developing a robust, end-to-end data processing pipeline (including chimera filtering, phasing, and fusion detection), (3) making the pipeline readily available, and by (4) testing the method on the largest targeted long-read (PacBio) sequencing cohort to date. A comparison of this study and previous studies is shown in Table 1. The samples tested in this study are from the Solomon Islands, a population that is largely uncharacterized for CYP2D6. The Solomon Islands are striving towards P. vivax

**Table 1 Comparison of long-read *CYP2D6* typing studies.**

| Study | Platform | Num. samples typed | Max. multiplex | Phasing | Chimera removal | Fusion calling | Copy num. | End-to-end pipeline available |
|---|---|---|---|---|---|---|---|---|
| Ammar et al.[27] | ONT MinION | 1 | 1 | Yes | No | No | qPCR | No |
| Qiao et al.[24] | PacBio RS II | 39 | NP | No[a] | No | No | qPCR | No |
| Buermans et al.[25] | PacBio RS II | 25 | 24 | Yes | Yes | PCR | PCR[b] | No |
| Liau et al.[26] | ONT GridION | 32 | 24 | Yes | No | No | PCR[b] | No |
| This study | PacBio Sequel I/II | 365 | 95 | Yes | Yes | PCR + Seq | qPCR | Yes |

*NP* not provided.
[a]Phasing performed for a subset of samples using additional allele-specific PCR, not long-read sequencing.
[b]Presence of duplication and deletion PCR products checked, however precise copy numbers of duplications were not attained.

**Table 2 Assigned *CYP2D6* alleles of control samples.**

| Sample | Reported alleles[a] | Assigned alleles | Assigned fusions | Copy num. | |
|---|---|---|---|---|---|
| | | | | intron-2 | exon-9 |
| NA07439 | *4×N,*41 | *4×2,*41 | – | 3 | 3 |
| NA10005 | *17,*29 | *17,*29 | – | 2 | 2 |
| NA12244 | *35,*41 | *35,*41 | – | 2 | 2 |
| NA17052 | *1,*1 | *1,*1 | – | 2 | 2 |
| NA17058 | *10,*10 | *10,*10 | CYP2D6-D7 (exon-8) | 4 | 2 |
| NA17203 | *4,*35 | *4,*35 | CYP2D6-D7 (intron-1) | 2 | 2 |
| NA17246 | *4,*35 | *4,*35 | CYP2D6-D7 (intron-1) | 2 | 2 |
| NA17252 | *4,*5 | *4,*5[b] | – | 1 | 1 |
| NA17280 | *2,*3 | *59,*3 | – | 2 | 2 |
| NA17300 | *1,*6 | *1,*6 | – | 2 | 2 |

[a]Reported consensus allele for Pratt et al.[14].
[b]Deletion allele (*5) inferred due to copy number 1 result from qPCR.

elimination and hence *CYP2D6* characterization is valuable for informing 8-aminoquinoline malaria pharmacotherapeutic strategies in this population. We elected to use the PacBio SMRT platform due to the low error rate of CCS reads. Our pipeline, Phased Long Allele Sequence Typing with Error Removal (PLASTER), is implemented in Nextflow[28] and available at https://github.com/bahlolab/PLASTER. The methods and pipeline presented here are broadly applicable to other polymorphic genes of interest in humans and other species.

## Results

**Copy number analysis.** *CYP2D6* copy number was assayed using probes targeting both intron-2 and exon-9, with a discordant result between the two probes suggesting the presence of a fusion allele (high intron-2 implies CYP2D6-D7 alleles, high exon-9 implies CYP2D7-D6 alleles). The performance of the qPCR assay was validated by the inclusion of ten control DNA samples with previously reported *CYP2D6* diplotypes[14] (Table 2 and Fig. 1b). Of the ten controls, nine were consistently assigned the expected copy number, including both a deletion control NA17252 (*4/*5) and a duplication control NA07439 (*4XN/*41) (Table 2 and Fig. 1b). Sample NA17058, with the previously assigned diplotype *10/*10, was typed as copy number 4 for intron-2 and copy number 2 for exon-9, indicating the presence of two copies of a CYP2D6-D7 allele. Notably, the *10 allele is reportedly frequently observed in a tandem arrangement with the CYP2D6-D7 fusion allele *36[29,30], which suggests that NA17058 may be harboring a *36-*10 tandem allele on each haplotype. From the Solomon Islands cohort, 377 samples were assayed of which 371 (98.4%) were successfully typed, with the remainder likely suffering from poor DNA quality. The most frequently assigned copy number state was two copies of both intron-2 and exon-9 (I2×2/E9×2) in 266 (71.7%) samples, followed by I2×3/

E9×2 (possible CYP2D6-D7 carriers) in 38 (10.2%), I2×3/E9×3 in 19 (5.1%), I2×1/E9×1 (*5) in 17 (4.6%), I2×2/E9×3 (possible CYP2D7-D6 carriers) in 14 (3.8%) and an additional five other copy number states making up the remaining 17 (4.5%) samples (Fig. 1a and Supplementary Data 4).

**Long-amplicon PCR.** *CYP2D6* was amplified with primers targeting a 6.1 kb region spanning from 712 bp upstream to 1176 bp downstream of the NG_008376.4 RefSeq CDS (Supplementary Fig. 1). By design, these primers also amplified the corresponding region from *CYP2D7*, generating a 7.6-kb product, to allow identification of *CYP2D6*/*CYP2D7* gene fusions (Supplementary Figs. 2 and 3). Pilot sequencing indicated that amplicons longer than 8 kb were not efficiently amplified and sequenced, so 8 kb was considered the upper limit for amplicon size. Both forward and reverse primers included adaptor sequences to facilitate downstream barcoding and multiplexed sequencing. Amplification was successful on both frozen whole blood (FWB) and dried blood spots (DBS) stored for 1 year prior to DNA extraction, where 89% of FWB (167/188) and 79% (31/39) of DBS amplified successfully, with no significant difference between the two groups (Fisher's exact test, $P = 0.12$, odds ratio = 2.04), noting that this test is underpowered with an ~32% chance to detect a significant difference at a threshold of 0.05.

To test the effects of storage times, FWB stored for 5 years prior to DNA extraction was tested and 90% (169/188) of samples were successfully amplified (Supplementary Data 1). This compared to 100% of DBS or FWB taken less than a week prior.

**Multiplexed amplicon SMRT sequencing.** Barcoded *CYP2D6* amplicons from a total of 377 samples and 10 controls were

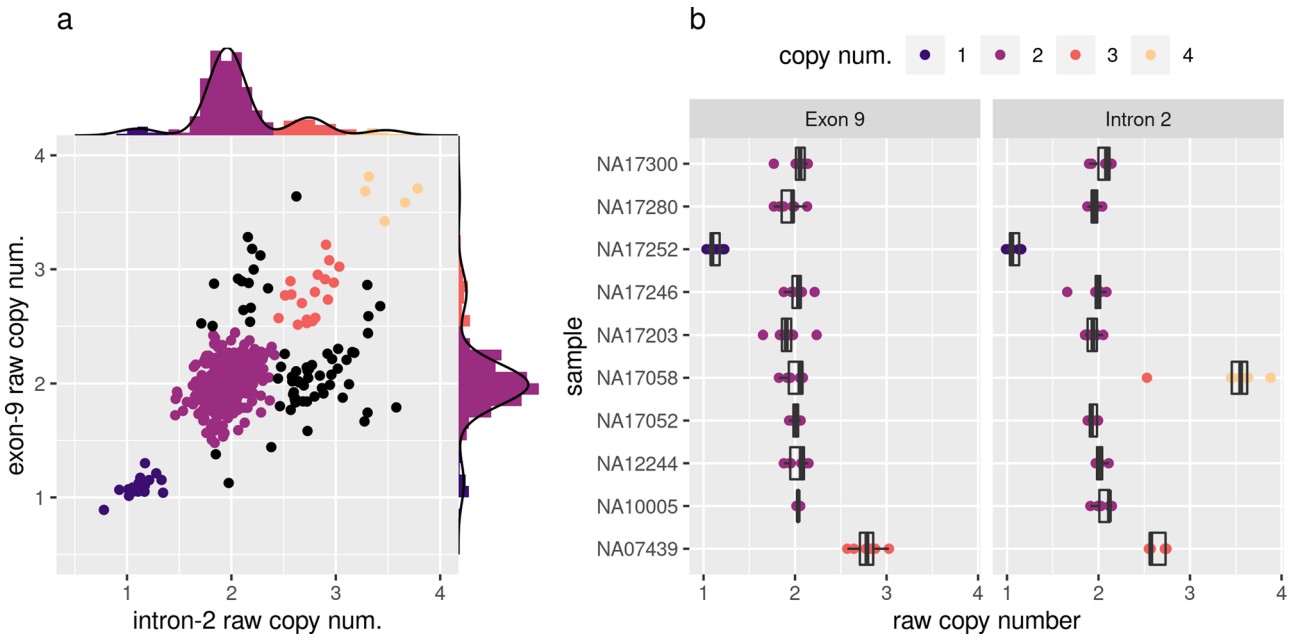

**Fig. 1 CYP2D6 intron-2 and exon-9 qPCR results. a** Scatterplot showing raw copy numbers for intron-2 and exon-9 qPCR assays for the Solomon Islands Cohort. Axis margins show histograms overlaid with the density from the Gaussian mixture models used to assign copy number states; **b** Tukey boxplots of raw copy number for control sample technical replicates repeated over several qPCR runs. Shown are seven technical replicates for all samples/markers except NA10005/Exon-9, NA17300/Exon-9, and NA17246/Intron-2 where only six were available, and NA12244/Intron-2 where only four were available.

sequenced over eight sequencing runs, with 30-95 multiplexed samples included in each run. These runs also included several other amplicons targeting additional genomic regions (data excluded from this analysis). Pre-processing of raw subreads was performed using the newly developed PLASTER pipeline, and across the eight runs an average of 75.1% (range 43.3–91.6%) of polymerase reads were mapped to the expected genomic loci (including non-CYP2D6 amplicons). For polymerase reads which mapped to CYP2D6/CYP2D7, an average of 15.8% (range 7.8–33.7%) of reads were retained after removing reads which failed to generate high-fidelity CCS reads, reads which did not have multiplexed barcode sequences intact, and reads which did not have the target sequence primers present and in the correct orientation (Supplementary Fig. 4 and Supplementary Data 3). The majority of CYP2D6/CYP2D7 reads mapped to CYP2D6 (mean 86.5%, range 84.1–89.8%), likely due to amplification bias as the CYP2D6 product is ~1.5 kbp shorter. A minimum coverage of 50 CYP2D6/CYP2D7 CCS after filtering was targeted, with an average of 74.0% (range 60.0–95.8%) of samples having sufficient coverage from a single run. In some instances, samples were included in multiple sequencing runs to attain sufficient coverage, leading to a total of 355/377 (94.2%) SI samples and 10/10 control samples with sufficient coverage for CYP2D6 allele typing.

***CYP2D6/CYP2D7 chimeras and gene fusions.*** CCS reads aligned to either CYP2D6 or CYP2D7 were screened for both PCR chimeras and genuine gene fusions between CYP2D6 and CYP2D7. Reads were designated as chimeric/fusion if they had better alignment to any potential single breakpoint CYP2D6-D7 or CYP2D7-D6 product derived from the hg38 reference sequences. Fusions were then called when a sufficient clustering of breakpoints was observed within a sample (see "Methods"), otherwise the chimeric/fusion reads were assumed to be PCR chimeras (Fig. 2 and Supplementary Fig. 5). Both fusion reads and chimeric reads were removed prior to downstream CYP2D6 allele typing. CYP2D6-D7 fusions were detected in three control samples, with samples NA17203 and NA17246 having a CYP2D6-D7 fusion with breakpoints in intron-1 consistent with

CYP2D6*68[31], and sample NA17058 having a CYP2D6-D7 fusion with the breakpoint in exon-8 consistent with CYP2D6*63[31] (Table 2). In the SI cohort, 27/355 (7.6%) samples were detected with a CYP2D6-D7 fusion allele with breakpoints in exon-8 (consistent with CYP2D6*63), and an additional 7/355 (2.0%) samples were detected with a CYP2D7-D6 fusion allele with breakpoints in intron-8 consistent with CYP2D6*13[11]. Discordant intron-2 and exon-9 copy numbers ascertained in the copy number assay were strongly associated with presence of fusion alleles. For samples with CYP2D6-D7 fusions calls (excluding those with a breakpoint before intron-2), 23/28 had high intron-2 counts versus 24/332 for those without (one-sided Fisher's exact test, odds ratio = 57.4, P value < $2.2 \times 10^{-16}$). For samples with CYP2D7-D6 fusion calls, 6/7 had high exon-9 counts vs 8/353 for those without (one-sided Fisher's exact test, odds ratio = 236.7, P value = $7.11 \times 10^{-9}$).

**Read phasing and genotyping.** Results from the copy number analysis were combined with the amplicon SMRT sequencing to improve the accuracy of CYP2D6 allele phasing by determining the maximum number of alleles (phases) to be detected during read phasing and chimera removal. This was conducted using a custom R package AmpPhaseR (Supplementary Fig. 6), which is part of the PLASTER pipeline. The number of alleles (phases) identified was one for 32.9% of samples, two for 66.0% of samples, and three for the remaining 1.1% of samples. For samples with one allele, a mean of 66.2% of reads were assigned to the allele and retained, while 20.7% and 13.2% were determined to be noise, or assigned to singletons respectively and removed. For samples with greater than one allele, a mean of 35.8% of reads were assigned to the allele and retained, while 12.4%, 9.4%, and 42.5% were determined to be chimera, noise, and singletons respectively and removed. Phased reads were then genotyped with GATK[32].

***CYP2D6 star allele typing.*** Ten control samples (20 alleles) with previously reported diplotypes were assayed[14], and of these 19/20 alleles were assigned the same star allele (Table 2). We

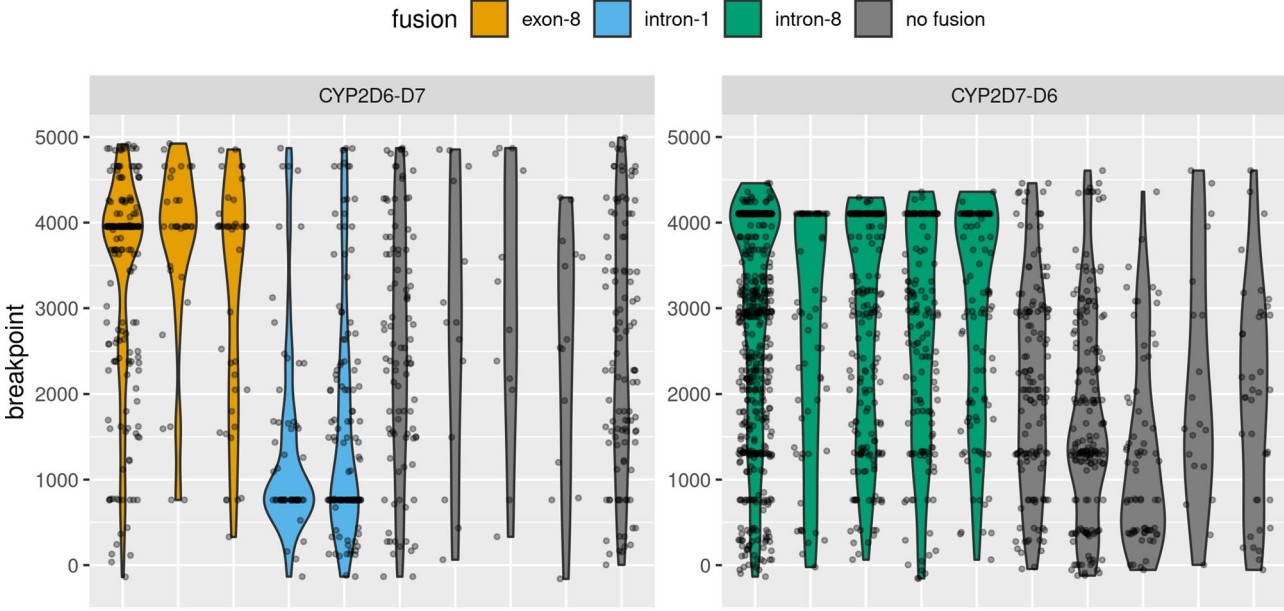

**Fig. 2 Fusion detection by breakpoint clustering.** Violin plots overlaid with jitter plots show the distribution of chimeric/fusion read breakpoints for a set of samples selected for illustrative purposes. Chimeric breakpoints are expected to a have a relatively uniform distribution over the length of the amplicon, while true gene fusions should be concentrated within a narrow region. Fusions are called when a sufficient density of breakpoints is detected (see "Methods" and source code for full details).

implemented a pipeline closely matching the methods employed by Buermans et al.[25] to compare the performance on control genotypes, and found a lower rate of concordance with 16/20 assigned the same star allele. The single discrepancy from our pipeline was from sample NA17280, which was originally reported as *2 but identified as *59 in our assay, which is an important distinction as *59 harbors the same set of SNPs as *2 in addition to a splice defect causing decreased function. Notably, 14/20 alleles were assigned to novel sub-alleles (not present in PharmVar) (Supplementary Data 4). A downsampling analysis was performed to determine the effect of CCS read depth on call rate and accuracy of the control sample genotypes. We found that performance started degrading marginally below 250 reads, decreasing to a call rate of 88% at 100 reads with 99% accuracy, and dropping to a call rate of 72% with 95% accuracy at 50 reads (Supplementary Fig. 7a). A similar analysis was performed for fusion allele typing, demonstrating that higher numbers of reads were required for consistent detection of fusion alleles (Supplementary Fig. 7b).

A total of 355 samples from the Solomon Islands population were allele typed with this pipeline, with 95.3% of sample alleles assigned to one of eight star alleles from the PharmVar catalog and the remaining 4.7% assigned to one of seven predicted novel star alleles with unknown function (Supplementary Data 4 and 5). Alleles were assigned to predicted novel star alleles based on the presence of variants with an impact assigned by VEP[33] as either "MODERATE" or "HIGH". The most frequently typed alleles were: *1 (68.0%, normal function), *10 (7.2%, decreased function), *71 (5.8%, uncertain function), *41 (5.5%, decreased function) and *2 (5.3%, normal function) (Fig. 3a). CYP2D6 activity scores were calculated for samples using the *CYP2D6* Allele Functionality Table available through the PharmGKB website. This resulted in 0.0% of samples being classified as PMs, 2.3% as IMs, 71.5% as EMs, 5.6% as UMs and the remaining 20.6% as unknown metabolizer status due to uncharacterized/novel *CYP2D6* alleles. However, in cases where not all sample alleles have a known activity score, a minimum activity score can still be calculated as the sum of all alleles with known activity scores, which resulted in

only 26.0% of unknown metabolizer status samples (5.3% overall) being possible PMs/IMs (Fig. 3b and Supplementary Data 6).

In addition to predicting the metabolizer status of samples, associations between fusion alleles, copy number states, and assigned star alleles were investigated using Fisher's exact tests. The CYP2D6-D7 exon-8 fusion (*CYP2D6 * 63*) was found to be strongly associated with the *10 allele (one-sided Fisher's exact test, odds ratio = 364.2, P value = $1.62 \times 10^{-25}$), suggesting the presence of a tandem arrangement. No other significant associations (P value < 0.01) were found between fusion allele presence and star alleles, or copy number states and star alleles (excluding *5).

## Discussion

We have developed a robust pipeline to sequence and phase alleles of the pharmacogenetically important gene *CYP2D6* using PacBio long-read amplicon sequencing. With rigorous filtering by read quality, PCR chimeras, and other sequencing artifacts we negate the effects of high-error rates and multiple PCR cycles and can identify rare and novel alleles in an uncharacterized population. We have tested the pipeline on a large cohort of samples collected from field sites in the Solomon Islands and identified alleles and defined haplotypes to predict CYP2D6 metabolizer status. The predicted CYP2D6 metabolizer status provides important information for malaria control in a population previously uncharacterized for CYP2D6 metabolizer status.

The benefits of using long-read sequencing include being able to identify known and unknown alleles, phasing of alleles, sequencing difficult regions, and identifying large structural variants. To test the method, we sequenced ten samples with known *CYP2D6* genotypes and successfully identified all short variants, phased alleles, and assigned *CYP2D6* star allele diplotypes as per nomenclature, as well as some additional allelic variants within the samples that have not been previously reported as far as we are aware. Copy number variants are difficult to ascertain from sequencing data alone. Therefore, we opted to use qPCR to identify copy number variants, which are important for many

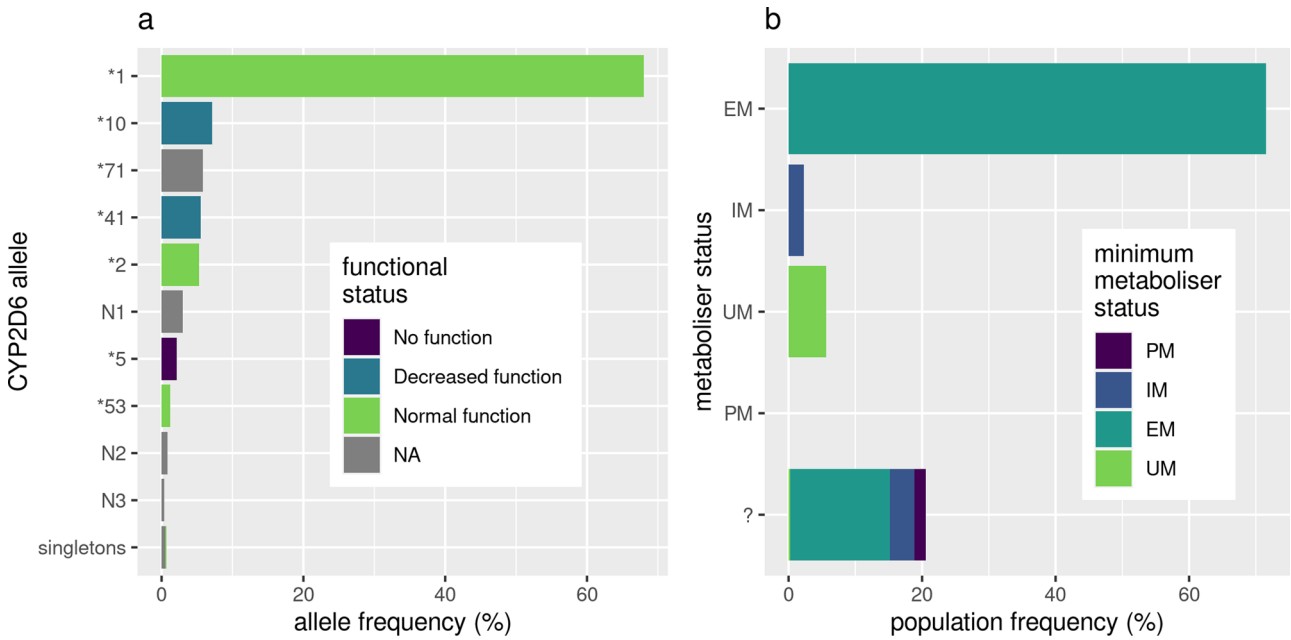

**Fig. 3 Frequency and activity of *CYP2D6* alleles in the Solomon Islands cohort. a** Frequency of *CYP2D6* alleles identified in the Solomon Islands cohort (*n* = 355), colored by *CYP2D6* functional status from the "*CYP2D6* Allele Functionality Table" (PharmGKB). Alleles prefixed with "N" represent novel alleles identified in this study that contain variants that are likely to impact protein function (VEP MODERATE and HIGH impact), full allele definitions are available in Supplementary Data 5. **b** Assigned metabolizer status of individuals in the Solomon Islands cohort (*n* = 355). In the case an individual contains one or more alleles with unknown function ("?"), the minimum metabolizer status is calculated based on the alleles present with known function.

treatments such as codeine (see guidelines by the clinical pharmacogenetics implementation consortium and other professional bodies[34]). Combining our pipeline with copy number analysis we identify many samples (including control samples previously reported *10 alleles) which are likely to be *10−*63 tandem duplications. The literature has reported *10 to commonly be in a tandem arrangement with *36 (CYP2D6-D7 with exon-9 breakpoint); however, we note that previous studies that have assigned the *36 allele based on qPCR alone, such as Del Tredici et al.[29], would not be able to distinguish between *36 and *63, and therefore it is unclear to what extent *63-*10 has been misclassified as *36-*10. This could be important as *36 is a nofunction allele, while *63 is an uncertain function allele. These insights on samples routinely used in *CYP2D6* studies demonstrate that our pipeline sets a standard whereby it can identify complex alleles that, until now, have remained concealed. In some cases these alleles may impact pharmacological outcomes, highlighting the benefit of improved *CYP2D6* analysis pipelines.

To facilitate future, rigorous long-read sequencing analysis with PacBio for complex fusion genes such as *CYP2D6*, we developed a set of guidelines for minimum data and quality required. We tested the method on long amplicons of 6–8 kb to maximize adaptability as these are frequently more difficult to sequence. Based on the downsampling analysis, a minimum of 50 high-quality CCS reads is recommended to phase and type nonfusion star alleles, with the best results being attained at 250 reads. For fusion alleles, a greater number of CCS reads was required, with a minimum of 200 reads being recommended and best results at 500 reads. For the Sequel I platform, the number of high-quality filtered CCS acquired across the seven runs was highly variable, with results between 13,000 and 85,000 (mean 41,000). Despite equimolar pooling based on quantification of double-stranded DNA, the number of CCS per sample-amplicon varied considerably and therefore, although theoretically 400 samples could be run simultaneously, we would suggest a maximum of 100 samples per run for the Sequel I platform. Our

dataset included a single Sequel II run, which yielded tenfold higher (440,000) high-quality filtered CCS, which would allow the maximum number of samples assayed on this platform to be increased.

The method described here is easily adaptable to other genes as a new gene of interest can be targeted by designing only the gene-specific primers with universal overhangs. The analysis pipeline can then be readily adapted to map the data to the region of interest and assign variants and phase the alleles. In addition, the cost to target additional genes is reduced by requiring only two gene-specific primers which must have a costly C6-amine block on them during PacBio sample preparation, as well as the ability to multiplex samples and genes in the same run. These factors in combination provide an adaptable method to analyze variants of other genes of interest in any population or species. For example, it would be beneficial to perform joint pharmacogenetic typing of *G6PD* along with *CYP2D6* when considering *Vivax* malaria control with primaquine, as G6PD deficiency can result in primaquine-induced hemolysis[35].

Drug safety and efficacy is of critical importance across the world, especially in areas with endemic infectious diseases requiring drug treatment. Local variations in pharmacogenetics could result in a high proportion of patients receiving ineffective treatments, essentially wasting a large portion of the limited budgets available for infection control in resource-poor areas. In addition, the local pharmacogenetic variants may place an unacceptably high proportion of individuals at risk from serious complications from treatment. Therefore, there is a need for a cost-effective method to screen populations in resource-poor areas for pharmacogenetic variants, prior to rollout of drug treatment programs.

To provide proof-of-concept that this method can be applied to samples obtained under field conditions with variable storage conditions, we utilized samples from a clinical trial that had been conducted in the Solomon Islands. Radical cure treatment for *P. vivax* malaria with primaquine is highly influenced by

pharmacogenetics as low CYP2D6 activity can lead to low amounts of the drug being metabolized and hence poor efficacy against the liver stage of *P. vivax*. We show that we can identify *CYP2D6* allele sets from either frozen blood or dried blood spots for the first time, to our knowledge, and from relatively old samples (1–5 years). Using the Activity Score (AS) system to predict phenotype from genotype we find 2.3–7.6% of this population are PM/IMs and therefore at risk of failed treatment and continuing to harbor *P. vivax*, potentially interfering with malaria elimination efforts. Over 20% of the population have uncharacterized or novel alleles demonstrating the diversity in *CYP2D6* globally, and the relative dearth of research in people not of European ancestry. These proportions could be substantially higher in other *P. vivax* endemic settings. This highlights the need for more genetic and matching phenotypic analyses of pharmacogenetics in these populations to better tailor treatment regimes.

The final limitation applicable to all pharmacogenetics is that genotype does not always easily translate to phenotype. The phased genotypic data provided by this pipeline is an important advance that could be linked to phenotypic analysis of samples to improve predictive capacity. Our results demonstrate that even *CYP2D6* control/reference samples thought to be well characterized still harbor missing information. We suggest that results from long-read sequencing should be used in the future to enhance the information contained in important pharmacological databases such as PharmVar.

There are many potential applications of this technology in the field of pharmacogenomics and beyond. Here it has been employed in a malaria-endemic area to predict the metabolism of PQ to an active drug, and this information could be used to help optimize PQ dosing to the population[36–38]. Determining the risk and effectiveness of drugs in populations may not only improve routine case management but could also contribute to the development of tailored public health campaigns, for example by aiding decisions between the use of mass-drug-administration or test-and-treat interventions[1,39,40]. This method could also be used to investigate the relationship between genotype and phenotype of pharmacogenetic variants as it has the capacity to identify all variants when used in combination with a copy number variant qPCR. Many common drugs including codeine are metabolized by CYP2D6[41] and this method could be used to enhance personalized medicine and improve drug safety. Individual pharmacogenetic profiles could be developed prior to attendance at a hospital. Overall, we show a robust, easily adaptable sequencing pipeline for genetic variants in field studies which we hope will lead to safer and more effective treatments.

## Methods

**Study samples**. The PCR was initially developed with human DNA purchased from Promega Australia (G1521). Comparisons between DNA extracted from Dried Blood Spots versus whole blood were performed with blood donations from the Victorian Blood Donor Registry (VBDR), Melbourne. Genotyping was tested with ten samples with a range of known *CYP2D6* genotypes[14] purchased from Coriell Institute for Medical Research.

Samples were collected less than a year prior to PacBio sequencing as part of a clinical trial of vivax radical cure in Tetere, Guadacanal Province Solomon Island (James, Karunajeewa and Mueller, personal communication/in prep). Individuals with a positive diagnosis of vivax malaria were recruited to the trial through local health care centers, and exclusion criteria were infancy (<1 year), pregnancy, a positive G6PD deficiency test or refusal or inability to provide informed written consent. Samples were collected as DBS (~50 µL) and whole blood from a finger-prick (~200 µL). DBS were dried and stored at room temperature (~30 °C) before being shipped to Melbourne, Australia. DNA was extracted within 6 months of sample collection using a genomic DNA blood extraction kit (Favorgen Biotech Corp, FADWE 96004). Whole blood was also collected, stored at −20 °C, and DNA extracted using the same kit. This study was approved by the Solomon Islands Health Research Ethics Review Board (HRE #041/16) and the Walter and Eliza Hall Institute Human Research Ethics Committee (HREC #16/02).

To test storage conditions and the effect on the amplification and sequencing protocols, whole blood was also tested from recent (<1 week) Melbourne, Australia

samples and from an older study conducted in Lihir, Papua New Guinea (Lautu, personal communication) where whole blood finger-prick samples had been collected 5 years prior and extracted two years prior to the PCR test and PacBio sequencing. After extraction, all DNA was stored at −20 °C for long-term storage, or at 4 °C for less than one month.

**Ethics**. Written informed consent was obtained from all study participants or their parents or legal guardians. This study was approved by the Solomon Islands Health Research Ethics Review Board (HRE #041/16) and the Walter and Eliza Hall Institute Human Research Ethics Committee (HREC #16/02).

**Quantitative PCR**. Quantitative PCR for *CYP2D6* copy number was performed via TaqMan™ Copy Number assays targeting Hs00010001_cn (exon-9) and Hs04083572_cn (intron-2) and run in a duplex with RNase P internal well controls. Each 11 µL reaction contained 5.5 µL TaqMan™ Fast Advanced Master Mix (Thermofisher Scientific, #4444557), 2 µL *CYP2D6* intron-2 or exon-9 TaqMan™ Copy Number Assay (FAM™ labeled, Thermofisher Scientific #4400291), 2 µL RNase P TaqMan™ Copy Number Reference Assay (VIC® labeled, Thermofisher Scientific #4403326), 2.4 µL nuclease-free water and 2 µL sample DNA solution. Assays were performed in 384 well plates, with each sample run with at least three technical replicates. Each plate contained ten control samples purchased from the Coriell Institute for Medical Research (NA07439, NA10005, NA12244, NA17052, NA17058, NA17203, NA17246, NA17252, and NA17280). Plates were run in a LightCycler® 480 Instrument (Roche) with a 10-min initial denaturation at 95 °C, followed by 40 cycles of 15 s denaturation at 95 °C and 1 min annealing/extension at 60 °C.

Fluorescence data were processed with LightCycler® 480 software (version 1.5.1.62), where $C_P$ (second derivative maximum) values were calculated and exported for downstream analysis. Copy number calls were obtained for each sample from the $C_P$ data using an in-house developed R Markdown analysis, available at https://github.com/bahlolab/PLASTER/. Briefly, raw Copy numbers were calculated from the $C_P$ values using the "delta-delta method"[42], wherein the median value across samples was assumed to be copy number 2 and used as the control. This normalization was performed separately for each plate and marker. Each sample-marker combination was then checked for outliers, before an average value was taken. Copy number calls were then assigned using a Gaussian mixture model. Software and packages used in this analysis were R 3.6.1[43], tidyverse packages (dplyr 1.0.2, ggplot2 3.3.2, magrittr 1.5, readr 1.3.1, tidyr 1.1.2)[44] and mixtools 1.2.0[45].

**Long-amplicon PCR**. Gene-specific primers were designed with Primer3 to amplify a 6.1-kb fragment, covering *CYP2D6* based on the hg38 reference genome, including the *CYP2D6* alternate haplotypes KB663609.1, KI270928.1, KN196486.1, KQ458388.1, and KQ759761.1. These primers also amplify a region of ~7.6 kb of *CYP2D7* in the hg38 reference genome, including alternate haplotypes GL383582.2, KI270928.1, KN196485.1, KN196486.1, KQ458387.1, KQ458388.1, and KQ759761.1 (Supplementary Fig. 1). The sample preparation flowchart is shown in Supplementary Fig. 8.

Primers were designed with a G/C end where possible to improve binding strength and up to three sets of primer pairs were tested. Primers that gave a strong specific band were then made with universal overhangs of 25 nucleotides appended to them and re-tested and only the strongest specific primer pairs were chosen as gene-specific primers. These gene-specific primers must include a C6-amine block to prevent unbarcoded amplicons from being sequenced. While this adds a significant cost to the PCR process, it is an essential step to gain the maximum number of barcoded reads.

Barcoding was performed in a second PCR, with a set of 96 barcodes appended to the forward and reverse universal overhangs. We used a modified PacBio reverse overhang and a custom forward overhang as it was found that using the PacBio forward and reverse universal overhangs resulted in non-specific products on the X chromosome when used with our gene-specific primers. We used the same barcode on the forward and reverse primers to aid with demultiplexing. The universal overhangs allowed a single set of universal barcoding primers to be used with any gene-specific primers significantly reducing the costs of multiplexing PCRs.

PCRs were performed with Takara PrimeSTAR GXL enzyme for 25 cycles with 3% DMSO (cat #R050B, Takara Bio). A subset of the DNA samples was quantified using QuantIT PicoGreen dsDNA assay kit (cat #P7589, Invitrogen) to determine the average DNA concentration in the samples. An average volume to contain 5 ng genomic DNA was amplified for 30 cycles with thermocycler conditions of denaturation at 98 °C for 3 min, cycling at 98 °C for 10 s, annealing 60 °C for 15 s, extension 68 °C for 6 min, final hold at 12 °C.

A subset of amplified samples was quantified using QuantIT PicoGreen dsDNA assay kit and the average volume to contain 10 ng of amplified DNA was taken into the next barcoding step and amplified for a further 26 cycles with the same conditions as before, except without the initial denaturation step. Barcode sequences were in forward or reverse complement orientation and matching pairs were used on each amplicon so that forward and reverse reads contained the same barcode. 96 barcode pairs were used thus the same amplicon from 96 samples could be sequenced in the same run, and multiple amplicons could be sequenced in

the same run. Theoretically, different barcodes could be used in forward and reverse and therefore more samples could be sequenced however with the current sequencing technology we calculated that more samples may not be sufficiently sequenced to gain enough high-quality reads (CCS reads) per amplicon. Forward and reverse barcode primer pairs were stored separately as we found evidence of self-annealing after a month in storage together.

PCR products were cleaned and size selected prior to pooling, to remove any primers and non-specific DNA to ensure accurate DNA quantification. Multiple methods could be used to achieve this. We used a magnetic bead-clean up protocol to remove small DNA and provide a level of size selection, using AMPure XP beads (Beckman Coulter (cat #100-265-900)) washed in molecular grade water 4 times and resuspended in the original buffer to make AMPure PB beads. We also tested the protocol using other magnetic bead purification kits with similar results. DNA was then quantified using PicoGreen dsDNA or Qubit HS dsDNA kit (ThermoFisher Scientific). Samples were then pooled equimolarly to give a final DNA amount of 1000 ng. In the absence of access to these kits size pooling could also be performed by running the samples on a gel, ensuring similar amounts of product added to a pool, and continuing with the first size selection step from the PacBio protocol immediately to remove short products and primer dimers. Immediately before proceeding to library preparation we checked the pooled sample for quality using a TapeStation (Agilent). A pooled sample only entered the library preparation step if a clear peak was visible with no tail of small products, indicating there was no DNA degradation.

**Sequencing**. Library preparation was performed as per the PacBio amplicon template preparation and sequencing protocol recommended for the reagents (PN 100-801-600 Version 04 and PN 101-791-800 version 01 (June 2019)). The sample was cleaned using magnetic bead separations with 0.45 volumes of AMPure PB beads (PacBio) after each step. Briefly, the protocol consisted of purifying the DNA, repairing damage such as nicks in the dsDNA, repairing DNA ends to form blunt-ended products, performing blunt-end ligation of the dsDNA product with SMRTbells, degrading any products not ligated to SMRTbells, and purifying the SMRTbell ligated products. The sequencing protocol was followed from the most recent SMRTlink version (SMRTlink v6) and consisted of annealing the sequencing primer, and the sequencing polymerase to the SMRTbells. All quality control steps were performed using TapeStation to assess the size of DNA, and Qubit High Sensitivity dsDNA assay to assess concentration.

The majority of the data were generated using Template Express kit 2.0, Sequencing primer v4, sequencing plate v3, and 1Mv3LR cells. Cells were loaded by diffusion loading with 4–6 picomolar of DNA with a 120-min immobilization and pre-extension step and a 20-h run time. Library preparation for run 8 was performed as per the PacBio amplicon template preparation and sequencing protocol (PN 101-791-800 version 01 (June 2019)). For runs 1–7, sequencing was performed using the PacBio Sequel machine at the WEHI Genomics Hub in Melbourne, Australia (PacBio Sequel CtrlVer 6.0.1.52258 and SigProcVer 6.0.0.47712). Run 8 was sequenced using the Sequel II platform at the Australian Genome Research Facility. On-plate loading concentration for this run was 70 pM, with 1.5 h of pre-extension and a movie time of 30 h.

**Data processing pipeline**. The entire data processing pipeline, PLASTER, has been implemented in Nextflow[28] and is available at https://github.com/bahlolab/PLASTER. The PacBio long-read amplicon sequencing data was processed in two stages: (1) pre-processing and (2) allele typing. Pre-processing begins with CCS generation from the subreads BAM file[46] (Signal-to-noise ratio (SNR) > 0.99, >=3 passes), followed by barcode demultiplexing with Lima[47]. The sequences are then aligned to the reference genome using pbmm2[48,49], and subsequently assigned to amplicons based on alignment position and the presence of primer sequences using several custom python scripts based on the pysam python library[50]. Reads without both sequencing primers present in the correct orientation are excluded at this point. Various run statistics are then collated and reported in an HTML document for quality control purposes. The primary output of this stage is a BAM file for each sample-amplicon combination.

The second stage, allele typing, begins with an optional fusion calling step which is targeted specifically for the detection of CYP2D6-D7 and CYP2D7-D6 fusions. Reads from the sample-amplicon BAM files from stage one assigned to either CYP2D6 or CYP2D7 are screened by multiple sequence alignment (MSA) to the corresponding CYP2D6 and CYP2D7 amplicons from the GRCh38 reference. This is then used to calculate an alignment score for each possible CYP2D6-D7 and CYP2D7-D6 fusion. If the highest fusion alignment score is at least ten base pairs greater than both the CYP2D6 and CYP2D7 alignment scores, the read is considered to be either the result of a PCR chimera or a fusion product. Gene fusion alleles are then detected by checking for a high density of predicted breakpoints around a single position (at least 25% of all fusion/chimeric reads within the sample). This stage is implemented in a custom R script ("fusion_call.R"), using the DECIPHER[51], RSamtools[52], and Biostrings[53] R packages, and is available as part of the Nextflow pipeline.

The allele-typing stage then continues by generating a preliminary set of diploid SNP calls with GATK HaplotypeCaller and GATK GenotypeGVCFs[32], following which SNP calls are assigned to individual reads using BCFtools[54]. A custom R

package AmpPhaseR was developed to perform phasing and chimera filtering of amplicon reads, incorporating external copy number information if available. SNP and indel variants are then re-called on phased reads with HaplotypeCaller and GenotypeGVCFs in haploid mode. Variant calls are then merged into a single VCF file for each amplicon, and variant annotation is performed using Ensembl VEP[33]. The choice to use GATK HaplotypeCaller and GenotypeGVCFs rather than a dedicated long-read caller was motivated by a publication by Wenger et al. wherein >99% precision, recall, and F1-score for SNVs using the well supported and validated GATK HaplotypeCaller on CCS reads was observed[21]. This obviates the need to use long-read specific callers, although this step could be replaced in the future.

We endeavored to benchmark our pipeline with that employed by Buermans et al. as this was the most compatible with our dataset of those listed in Table 1. It was not possible to run the pipeline exactly as implemented by Buermans et al. because the version of the PacBio SMRT portal (v2.3.0) long-amplicon analysis used is compatible only with PacBio RS II data, not the more recent PacBio Sequel data of our study. In addition, the Buermans et al. dataset did not include multiple amplicons as our dataset does and consequently has no stage for splitting amplicons. Nonetheless, we implemented a Nextflow pipeline matching the methods used as closely as possible, using a current version of PacBio's long-amplicon analysis (2.4.2) and the amplicon splitting stage from PLASTER, and compared the resulting CYP2D6 star alleles assignments of the control samples. This benchmarking pipeline is available at https://github.com/bahlolab/cyp2d6-laa-benchmark.

**Assignment of *CYP2D6* star alleles**. Star alleles were assigned to phased variants by comparison to those provided in VCF format in the PharmVar CYP2D6 Gene Data Download (v4.2.6.1). In order to be assigned a given star allele, all variants present in the PharmVar reference allele must be present in the sample-phase variant set. In the case that multiple star alleles were matched using this method, ties were broken first by selecting the allele with the lowest activity score, second by selecting the core allele with the greatest number of variants, and third by then selecting the allele with the lowest star number. This method is based on criteria described on the PharmVar website. Note that due to the region targeted by sequencing primers the following sets of CYP2D6 sub-alleles are not able to be distinguished; (1) *2.001, *2.011, *2.019 and *2.029; (2) *6.005 and *6.006; and (3) *71.001 and *71.003. This is implemented in a custom R script ("pharmvar_star_allele.R") and performed as an optional final step in the PLASTER allele-typing pipeline, noting that other pharmacogenes in the PharmVar database are also supported. Activity scores were obtained from the CYP2D6 Allele Functionality table available on the PharmGKB website (https://www.pharmgkb.org/page/cyp2d6RefMaterials).

**Downsampling analysis**. A downsampling analysis was performed in order to determine the effect of CCS read number on the call rate and accuracy of CYP2D6 genotypes. For this analysis, a subset of the samples was used, which was comprised of the 10 control samples and 40 randomly selected Solomon Islands samples each with greater than 500 filtered CYP2D6 CCS reads. Five independent replicate downsampling runs were performed at read numbers of 500, 400, 300, then at intervals of 10 from 250 down to 10 reads. This analysis only used CCS reads assigned to CYP2D6 (not CYP2D7). It was noted that all replicates of the controls agreed with previously reported allele types at the 500 read depth, so allele types assigned at this read depth were used as the truth set for calculating accuracy. Accuracy was assigned at the level of matching CYP2D6 core star alleles (not sub-alleles or individual variants). Of the Solomon Islands cohort samples included, three were assigned novel allele types, which are not consistently assigned the same name by the pipeline, and as such were excluded from the downsampling analysis leaving a total of 47 samples. A second downsampling analysis was performed to determine the effect of CCS read number on the accuracy of CYP2D6-D7 fusion alleles. For this analysis, the 20 samples (including three control samples) with detected fusion alleles and greater than 500 CYP2D6/CYP2D7 CCS reads were used. Five independent replicate downsampling runs were performed at read numbers of 500 down to 25 at intervals of 25. Accuracy was determined based on matching the consensus fusion allele typed at 500 reads, with breakpoints allowed to differ up to 20 bp from the consensus breakpoint. Call rate and accuracy were not able to be independently assessed as no-call was equivalent to no-fusion for this portion of the pipeline.

**Statistics and reproducibility**. Statistical tests such as Fisher's exact tests are described in full in the relevant results sections.

**Reporting summary**. Further information on research design is available in the Nature Research Reporting Summary linked to this article.

## Data availability

The PacBio CCS data (output from the PLASTER pre-processing stage) is available through the NCBI Sequence Read Archive (SRA) (PRJNA754842). Raw qPCR data are available in the PLASTER GitHub repository (https://github.com/bahlolab/PLASTER).

Source data underlying figures are presented in Supplementary Data 7. All other data are available from the corresponding author on reasonable request.

## Code availability

The PLASTER pipeline software is available at https://github.com/bahlolab/PLASTER and released under the MIT license.

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

## Acknowledgements

We thank the participants, their families and carers, and the community of the Tetere region of Guadalcanal for providing the samples used in this study. We also thank Albino Bobogare from the Solomon Islands Ministry of Health and Medical Services, the team of research field workers, and the health services of the Good Samaritan Hospital in Tetere who made this work possible. The authors wish to acknowledge the use of the services and facilities of both the WEHI Genomics Hub and the Australian Genome Research Facility. This work was supported by an NIH International Centre of Excellence for Malaria Research (ICEMR) Grant

(RFA-AI-15-056), the National Health & Medical Research Council (NHMRC) (GNT1092789), and the Bill & Melinda Gates Foundation (OPP1151132). I.M. was supported by an NHMRC Principal Research Fellowship (GNT1155075). M.B. was supported by a NHMRC Investigator Grant (GNT1195236). This work was also supported by the Victorian Government's Operational Infrastructure Support Program and the NHMRC Independent Research Institute Infrastructure Support Scheme (IRIISS).

## Author contributions

S.C. conceived the project and developed the amplicon sequencing strategy. J.M. developed the data processing pipeline and performed statistical and bioinformatic analyses. S.C., L.S., J.B., C.B., and S.R.P. prepared samples for sequencing. R.M. performed the qPCR experiments. R.J., H.K., and I.M. along with Albino Bobogare were responsible for the clinical trial from which the Solomon Islands samples were attained. D.L.G. worked on the study from which the Lihir samples were attained. I.M. and M.B. supervised the project. J.M. and S.C. wrote the manuscript with input from all authors.

## Competing interests

M.B. is an Editorial Board Member for *Communications Biology*, but was neither involved in the editorial review nor the decision to publish this article. The remaining authors declare no competing interests.
