## [Peer Review File · Communications Biology]

Reviewers' comments:

Reviewer #1 (Remarks to the Author):

The authors describe a pipeline for accurate allelotyping of the CYP2D6 gene using long read amplicons. CYP2D6 is an important locus due to its relevance in metabolism of drugs and the presence of a pseudo-gene complicates the process of sequencing this locus. There have been many published papers on genotyping CYP2D6 using long read sequencing technologies (as summarized by the authors in Table 1). The strengths of this paper - as summarized by the authors - are a robust data processing pipeline and the large number of samples that they genotyped. Overall, the paper is well written and easy to follow. Since this problem has been tackled by several groups, I think that the authors could have done additional analysis in order to distinguish their work from previously published work and demonstrate the broad utility and robustness of their pipeline.

1. There have been several pipelines developed for genotyping CYP2D6 from Illumina WGS data (e.g. Cyrius, <https://www.nature.com/articles/s41397-020-00205-5>).

How does the targeted long read based sequencing compare to the Illumina based allelotyping? One advantage of the Illumina WGS based genotyping is that copy number does not need to be measured separately using qPCR.

2. In Table 1, the authors list previous studies for CYP2D6 typing. The Buermans et al. study is listed as having the pipeline available. It would be useful to analyze the data generated in this study using this pipeline for direct comparison.

3. Along the same lines as (2) above, the authors could analyze previously published targeted long read data for CYP2D6 to demonstrate that their pipeline works for different types of long read amplicon data and sequencing technologies.

Reviewer #2 (Remarks to the Author):

Dear authors,

I wish to congratulate the authors on this interesting work. The manuscript is well written and the experiments well described and executed. The application of long-read sequencing to these challenging parts of the genome is highly suitable. I have only minor comments.

Sincerely,

Wouter De Coster

Page 4, line 107 mentions the accuracy of long-read sequencing platforms. These error rates are, at least for ONT, highly outdated, and 2D consensus sequencing has also been deprecated since 2017. The company claims the modal accuracy of the latest base callers and chemistry is at 99%, but conservatively ~5% would be a reasonable estimate of currently produced data. Finding citable references for such a moving target is challenging, but <https://www.nature.com/articles/s41588-021-00865-4> reports a median accuracy of 11.6%. Better results are obtained with the 1D² method: <https://genomebiology.biomedcentral.com/articles/10.1186/s13059-020-02255-1>

On page 14, line 337 recommendations are made for the number of reads that are necessary for the identification of fusion alleles and variant phasing. It is however not clear to me how these numbers were determined, and if a downsampling analysis was performed to obtain these estimates.

The authors used GATK HaplotypeCaller for variant calling, which may be appropriate for high-quality HiFi reads, but I wonder about the reason to not use a variant caller tailored to long-read sequencing, e.g. DeepVariant, Clair, or Longshot? While this does probably not affect the results in a major way, some of these tools additionally leverage the phasing of variants or are more appropriate for the error types in long-read sequencing, and I believe it would be relevant to comment on the choice of variant caller.

Reviewer #1 (Remarks to the Author):

The authors describe a pipeline for accurate allelotyping of the CYP2D6 gene using long read amplicons. CYP2D6 is an important locus due to its relevance in metabolism of drugs and the presence of a pseudo-gene complicates the process of sequencing this locus. There have been many published papers on genotyping CYP2D6 using long read sequencing technologies (as summarized by the authors in Table 1). The strengths of this paper - as summarized by the authors - are a robust data processing pipeline and the large number of samples that they genotyped. Overall, the paper is well written and easy to follow. Since this problem has been tackled by several groups, I think that the authors could have done additional analysis in order to distinguish their work from previously published work and demonstrate the broad utility and robustness of their pipeline.

1. There have been several pipelines developed for genotyping CYP2D6 from Illumina WGS data (e.g. Cyrius, <https://www.nature.com/articles/s41397-020-00205-5>). How does the targeted long read based sequencing compare to the Illumina based allelotyping? One advantage of the Illumina WGS based genotyping is that copy number does not need to be measured separately using qPCR.
 - We thank the reviewer for this comment, and we agree that Cyrius in particular appears to be a good approach to genotyping CYP2D6 genotypes from short-read WGS. While Cyrius has the advantage of not requiring a separate qPCR assay, we note that Cyrius does not have any means of typing novel CYP2D6 alleles, and running short-read WGS in place of multiplexed long-read amplicon sequencing for CYP2D6 genotypes alone is not cost effective. We have added this information and the reference to Cyrius in the introduction (line 96-98).
2. In Table 1, the authors list previous studies for CYP2D6 typing. The Buermans et al. study is listed as having the pipeline available. It would be useful to analyze the data generated in this study using this pipeline for direct comparison.
 - We thank the reviewer for this comment. The “pipeline” used in the Buermans et al. study we noted as being available is PacBio’s long amplicon analysis protocol, and only carries out a portion of the analysis steps that are part of our much more comprehensive pipeline (essentially clustering/phasing and chimera removal, but not allele-typing). We have updated Table 1 with the column “End-to-end Pipeline available” to make it clearer that no previous publications have released an end-to-end pipeline. We do agree that it would be useful to have a benchmark of a pipeline used by another study, so we have implemented as close as possible the complete pipeline used by Buermans et al (which is only a subset of our more complete end to end pipeline) and have tested our data using this pipeline. We have updated the results section at lines 261-264 reflecting this benchmark, as well as the methods section at lines 553-562.
3. Along the same lines as (2) above, the authors could analyze previously published targeted long read data for CYP2D6 to demonstrate that their pipeline works for different types of long read amplicon data and sequencing technologies.
 - We thank the reviewer for this comment. The pipeline as it exists now is specific to PacBio data. In particular the pre-processing stage uses PacBio specific tools (ccs, lima). However, the pipeline has been released as free and open-source software, and it would

be feasible for a separate pre-processing module to be written for other sequencing technologies. Additionally, no other study to our knowledge has publicly released PacBio CYP2D6 sequencing data.

Reviewer #2 (Remarks to the Author):

Dear authors,

I wish to congratulate the authors on this interesting work. The manuscript is well written and the experiments well described and executed. The application of long-read sequencing to these challenging parts of the genome is highly suitable. I have only minor comments.

Sincerely,

Wouter De Coster

1. Page 4, line 107 mentions the accuracy of long-read sequencing platforms. These error rates are, at least for ONT, highly outdated, and 2D consensus sequencing has also been deprecated since 2017. The company claims the modal accuracy of the latest base callers and chemistry is at 99%, but conservatively ~5% would be a reasonable estimate of currently produced data. Finding citable references for such a moving target is challenging, but <https://www.nature.com/articles/s41588-021-00865-4> reports a median accuracy of 11.6%. Better results are obtained with the 1D² method: <https://genomebiology.biomedcentral.com/articles/10.1186/s13059-020-02255-1>
 - We thank the reviewer for this comment and acknowledge that the 2017 benchmarking paper is outdated, however the strength of this reference is that it compares the two platforms directly using matched methodology. We have added more recent estimates to lines 111-112.
2. On page 14, line 337 recommendations are made for the number of reads that are necessary for the identification of fusion alleles and variant phasing. It is however not clear to me how these numbers were determined, and if a downsampling analysis was performed to obtain these estimates.
 - We thank the reviewer for this comment and agree that a formal downsampling analysis would be appropriate to support these recommendations. This has now been performed and the results have been included in the manuscript in lines 267-273 of the results section, lines 351-355 of the discussion and lines 581-601 of the methods section as well as being reflected with a new supplemental figure S6.
3. The authors used GATK HaplotypeCaller for variant calling, which may be appropriate for high-quality HiFi reads, but I wonder about the reason to not use a variant caller tailored to long-read sequencing, e.g. DeepVariant, Clair, or Longshot? While this does probably not affect the results in a major way, some of these tools additionally leverage the phasing of variants or are more appropriate for the error types in long-read sequencing, and I believe it would be relevant to comment on the choice of variant caller.
 - We thank the reviewer for this comment, and have added a note in the methods section (lines 549-551) regarding the choice of variant caller. The choice was made due to the demonstrated performance of HaplotypeCaller on HiFi data and our own familiarity with the tool.

REVIEWERS' COMMENTS:

Reviewer #1 (Remarks to the Author):

The authors have addressed the comments from the previous review in the revised manuscript. I have no further comments.

Reviewer #2 (Remarks to the Author):

The authors have adequately answered my comments. I have no further comments and I recommend this manuscript for publication.

Sincerely,
Wouter De Coster